# Artificial Intelligence Model Assists Knee Osteoarthritis Diagnosis via Determination of K-L Grade

**DOI:** 10.3390/diagnostics15101220

**Published:** 2025-05-12

**Authors:** Joo Chan Choi, Min Young Jeong, Young Jae Kim, Kwang Gi Kim

**Affiliations:** 1Department of Biomedical Engineering, College of Health & Science, Gachon University, Seongnam-si 461701, Republic of Korea; 2Department of Electrical and Computer Engineering, College of Information and Communication Engineering, Sungkyunkwan University, Suwon 16419, Republic of Korea; 3Gachon Biomedical & Convergence Institute, Gachon University Gil Medical Center, Incheon 21565, Republic of Korea; 4Deptartment of Biomedical Engineering, College of Medicine, Gachon University Gil Medical Center, 38-13 3beon-gil, Namdong-gu, Incheon 21565, Republic of Korea; 5Deptartment of Health Science & Technology, Gachon Advanced Institute for Health Science & Technology (GAIHIST), Lee Gil Ya Cancer and Diabetes Institute, Gachon University, 155 Gaetbeol-ro, Yeonsu-gu, Incheon 21999, Republic of Korea

**Keywords:** knee osteoarthritis, Kellgren–Lawrence grading system, deep learning, DenseNet201, ResNet101, EfficientNetV2

## Abstract

**Background:** Knee osteoarthritis (KOA) affects 37% of individuals aged ≥ 60 years in the national health survey, causing pain, discomfort, and reduced functional independence. **Methods:** This study aims to automate the assessment of KOA severity by training deep learning models using the Kellgren–Lawrence grading system (class 0~4). A total of 15,000 images were used, with 3000 images collected for each grade. The learning models utilized were DenseNet201, ResNet101, and EfficientNetV2, and their performance in lesion classification was evaluated and compared. Statistical metrics, including accuracy, precision, recall, and F1-score, were employed to assess the feasibility of applying deep learning models for KOA classification. **Results:** Among these four metrics, DenseNet201 achieved the highest performance, while the ResNet101 model recorded the lowest. DenseNet201 demonstrated the best performance with an overall accuracy of 73%. The model’s accuracy by K-L grade was 80.7% for K-L Grade 0, 53.7% for K-L Grade 1, 72.7% for K-L Grade 2, 75.3% for K-L Grade 3, and 82.7% for K-L Grade 4. The model achieved a precision of 73.2%, a recall of 73%, and an F1-score of 72.7%. **Conclusions:** These results highlight the potential of deep learning models for assisting specialists in diagnosing the severity of KOA by automatically assigning K-L grades to patient data.

## 1. Introduction

Knee osteoarthritis (KOA) is a leading cause of disability among both older and younger adults. As the global population ages, the number of people affected by KOA is projected to reach 130 million people by 2050 [1]. Radiographic grading for diagnosing osteoarthritis (OA) primarily relies on the Kellgren–Lawrence (K-L) grading system, which examines changes visible in plain radiographs, such as X-rays [2]. This system assesses joint space narrowing, osteophyte formation, and subchondral sclerosis, grading severity from 0 to 4. The grading process is generally overseen by a specialist, and its accuracy is largely dependent on the specialist’s level of expertise, making it inherently subjective. As a result, the physician may assign different K-L grades when evaluating the same knee joint at different times [3]. In a 2015 study by Culvernor et al., intra-rater reliability for K-L grading ranged from 0.67 to 0.73 [4]. Computer-aided diagnosis can mitigate this subjectivity by providing automated assessments [5]. Notably, convolutional neural networks (CNNs) have shown comparable performance to arthroplasty specialists when evaluating KOA severity using the intraclass correlation coefficient (ICC) as a benchmark [6]. In recent years, deep-learning-based methods, such as CNNs, are being increasingly used for automatic diagnosis radiographic images [7].

In 2018, Aleksei Tiulpin et al. conducted a study to classify the severity of KOA based on the K-L grade. They trained a model based on ResNet34 using 18,376 images from the MOST public dataset provided by public institutions and tested it on the Osteoarthritis Initiative (OAI) dataset, achieving a performance of >66% [8]. In 2016, Antony et al. utilized approximately 2200 baseline images from the OAI and Multicenter Osteoarthritis Study (MOST) cohorts to train and test a CNN-based fine-tuned model for K-L grade classification, achieving a performance of over 57% [9]. In 2020, Wang et al. trained and tested a ResNet50 model using 4796 images from the OAI dataset for K-L grade classification, achieving an accuracy of approximately 69.18%, which was 2.5 percentage points higher than the baseline model [10]. In 2021, Olsson et al. conducted training and testing using a CNN model for K-L grade classification, achieving a mean AUC of 0.92 for all grades except K-L Grade 2 [11]. In 2021, Cheung et al. compared the effectiveness of CNNs in predicting the severity and progression of KOA. Using 4216 images from the Osteoarthritis Initiative (OAI) public dataset, they performed segmentation of the joint space width (JSW) and classified the K-L grade using an XGBoost model, achieving an AUC of 0.621 [12]. In 2022, Chern et al. classified the severity of KOA using 5000 images from the Osteoarthritis Initiative public dataset, with 1000 images per K-L grade. The authors combined DenseNet201 with a Support Vector Machine (SVM), achieving a maximum AUC performance of 71.33% [13].

Previous studies have primarily used public datasets rather than clinical data, employing CNN-architecture-based deep learning models to classify the severity of KOA. However, this approach is limited by imbalanced data across K-L grades. In this study, three deep learning models (EfficientNetV2, DenseNet201, and ResNet101) were trained and evaluated using 15,000 clinical images (3000 images per K-L grade). The models were assessed for both quantitative performance and qualitative performance to identify the model with the best performance in K-L grade classification.

Therefore, the main contributions of this study are as follows:(1)We constructed a balanced and clinically relevant KOA dataset consisting of 15,000 knee X-ray images, with 3000 images per K-L grade, collected from real patients in a hospital setting.(2)Using this clinical dataset, we comparatively evaluated the classification performance of three widely used deep learning architectures (EfficientNetV2, DenseNet201, and ResNet101) for automatic K-L grading.(3)In addition to quantitative performance metrics, we conducted qualitative analysis using Grad-CAM visualizations to examine the interpretability of the models.(4)Based on the experimental findings, we propose directions for future research, including feature map fusion from multiple architectures and ensemble model development to further enhance classification performance.

## 2. Materials and Methods

### 2.1. Data Collection

Knee anteroposterior (AP) X-ray data were collected from patients who underwent KOA examinations at the Catholic University St. Mary’s Hospital. The study received approval from the Institutional Review Board (IRB) of the Catholic University St. Mary’s Hospital (approval no. KC23RIDI0485). All methods were performed in accordance with the relevant guidelines and regulations, including the Declaration of Helsinki. As this was a retrospective study, informed consent was waived by the IRB. The K-L grading system classifies the severity of KOA into five grades (Figure 1), where “grade 0” indicates a normal knee, and “grades 1 to 4” represent increasing stages of OA progression.

All data used were labeled by at least two radiologists, who designated regions of interest (ROIs) within the images. Annotations were determined through consensus among multiple specialists and subsequently used in the study. The ROIs were set to include the knee osteophyte areas, highlighting features such as joint space narrowing, osteophytes, and cartilage loss in the training images. To prevent data imbalance, 3000 images were collected for each K-L grade (grade 0 to 4). From the dataset of 15,000 images, 12,000 were allocated for training, 1500 for validation, and 1500 for testing. The overall research workflow is illustrated in Figure 2.

### 2.2. Experimental Environment

This study was conducted on a system with an x86_64 processor and 251.49 GB of RAM, running on the Linux 6.5.0-41-generic operating system. The experiments were performed using Python (version 3.9.16). For image preprocessing and deep learning training, the following libraries were used: TensorFlow (Version 2.11.0), Keras (Version 2.11.0), OpenCV (Version 4.7.0), and PyTorch (Version 2.4.0 + cu121). The deep learning training was conducted using a single NVIDIA RTX A5000 24GB (NVIDIA, Santa Clara, CA, USA) GPU, operating in a CUDA (version 12.1) environment.

### 2.3. K-L Grade Classification Models

For K-L grade classification, the models selected were EfficientNetV2, DenseNet201, and ResNet101, all of which were trained using transfer learning with pre-trained weights and evaluated on the same dataset.

In selecting models for K-L grade classification, we carefully considered the unique characteristics of KOA medical imaging. Accurate KOA severity assessment requires the detection of subtle and complex pathological changes, such as joint space narrowing, osteophyte formation, and cartilage degradation. DenseNet201 was selected for its dense connectivity mechanism, which facilitates efficient feature reuse and the extraction of fine-grained details critical for identifying early-stage lesions. ResNet101 was chosen for its ability to maintain stable learning in very deep networks through residual connections, effectively modeling the hierarchical and complex anatomical structures of the knee joint. EfficientNetV2 was adopted for its computational efficiency and strong generalization capabilities, achieved through progressive learning and Fused-MBConv blocks, making it particularly suitable for handling relatively limited but high-quality medical datasets. Each architecture offers distinct advantages in terms of feature extraction, training stability, and classification performance, making them individually well suited for the task of KOA severity grading.

The EfficientNetV2 model, an improved version of the original EfficientNet, uses Fused-MBConv blocks for more efficient feature extraction. Fused-MBConv optimizes performance by using standard convolution for low-resolution images and MBConv for high-resolution images, enabling faster training. Additionally, it employs a Progressive Learning technique, starting with low-resolution images at the initial stages of training and gradually moving to more complex high-resolution images, thus reducing computational cost while enhancing accuracy [14].

The DenseNet201 model is a variant of the DenseNet (Dense Convolutional Network) architecture, where each layer receives input from all previous layers, facilitating efficient feature reuse and improved information flow. DenseNet201 has a total of 201 layers and strengthens inter-layer connections, addressing the vanishing gradient problem and enhancing learning efficiency. By connecting feature maps between blocks, DenseNet201 maintains high performance with fewer parameters, maximizing the model’s efficiency through feature reuse [15].

The ResNet101 model addresses the vanishing gradient problem in deep networks through residual connections, allowing each layer to reuse the output of previous layers, thus enabling efficient learning. This model also enhances critical lesion features and reduces the impact of background noise through deep and multi-scale feature extraction modules, resulting in more accurate image classification [16].

For all three models, CrossEntropyLoss was used as the loss function. The training environment was configured with a learning rate of 1 × 10^−4^, 50 epochs, and a batch size of 8. To optimize the model parameters and prevent overfitting, a checkpoint strategy was adopted to save only the model weights corresponding to the best validation loss. Additionally, a learning rate scheduler was employed to adjust the learning rate during training. Data augmentation techniques were not applied in this study.

### 2.4. Model Evaluation

The performance evaluation metrics for the classification models included accuracy, precision, recall, F1-score, Receiver Operating Characteristic (ROC) Curve, and Area Under the Curve (AUC), as summarized with their formulas in Table 1. To identify feature importance in the K-L grade predictions of the three models, Grad-CAM was used to visualize the areas of the images that influenced the model’s predictions.

## 3. Results

The automatic K-L grade classification model was trained using the AP X-ray data of patients who underwent KOA examination and the regions of interest (ROIs) designated by specialists. To determine whether each classification model accurately classified the K-L grades, class-specific metrics, including accuracy, precision, recall, F1-score, ROC Curve, and Area Under the Curve (AUC), were calculated, and the performance of the models was compared and analyzed. Table 2 presents the performance of the three deep learning models—DenseNet201, ResNet101, and EfficientNetV2—based on five types of performance metrics: accuracy, precision, recall, F1-score, and AUROC.

To further assess the learning behavior and generalization ability of the models during training, the changes in training loss and validation loss across epochs were plotted, as shown in Figure 3. Although slight overfitting was observed in the later epochs for some models, early stopping and checkpoint strategies based on validation performance were applied to prevent overfitting effectively.

DenseNet201 had the highest performance across all four key metrics. By contrast, the performance of ResNet101 ranked the lowest. DenseNet201 achieved approximately 73% accuracy, 73.2% precision, 73% recall, and an F1-score of 72.7%, demonstrating the most robust performance. ResNet101 had 69.6% accuracy, 69.7% precision, 69.6% recall, and an F1-score of 69.3%, while EfficientNetV2 achieved around 71.8% accuracy, 72% precision, 71.8% recall, and an F1-score of 71.6%. In terms of the AUC (%), EfficientNetV2 showed the highest performance (94.3%), followed closely by DenseNet201 (94.1%). Correspondingly, ResNet101 showed the lowest AUC performance (92.8%). The results for these three models are presented using a confusion matrix and AUROC curve in Figure 4 and Figure 5, respectively.

To analyze the classification performance of the three models more precisely, we visualized the classification accuracy for each K-L grade, as shown in Table 3.

For the DenseNet201 model, the classification accuracies were as follows: 80.67% for K-L Grade 0, 53.67% for K-L Grade 1, 72.67% for K-L Grade 2, 75.33% for K-L Grade 3, and 82.67% for K-L Grade 4.

The EfficientNetV2 model achieved accuracies of 76.33% for K-L Grade 0, 51.33% for K-L Grade 1, 73.67% for K-L Grade 2, 78.00% for K-L Grade 3, and 80.00% for K-L Grade 4.

The ResNet101 model showed the lowest performance among the three, with accuracies of 77.67% for K-L Grade 0, 49.00% for K-L Grade 1, 68.00% for K-L Grade 2, 71.67% for K-L Grade 3, and 82.00% for K-L Grade 4.

All three models demonstrated the ability to effectively classify higher severity cases, specifically K-L Grades 3 and 4. However, they faced notable challenges when classifying K-L Grade 1, indicating a common difficulty in distinguishing mild osteoarthritis cases.

To identify the specific regions of the image that influenced the classification results, Grad-CAM was used for visualization [17]. For the same image, attention maps were generated from the final layers just before classification by each of the three models—DenseNet201, ResNet101, and EfficientNetV2—and are shown in Figure 6. The results revealed that EfficientNetV2 and ResNet101, which recorded accuracies in the 69% to 71% range, either showed several regions influencing the classification beyond the osteophytes on both sides of the knee displaying joint space narrowing or displayed attention maps with overly small regions of interest. In contrast, DenseNet201, which achieved an accuracy in the 73% range, demonstrated well-localized attention focused primarily on the knee osteophytes showing joint space narrowing.

## 4. Discussion

This study aimed to mitigate the subjectivity inherent to the process used by specialists to perform their diagnoses. Our approach involved enabling an automated evaluation of the KOA K-L grades using deep-learning-based classification models in computer-aided diagnosis. To evaluate the effectiveness of this approach, the performance of three models—DenseNet201, ResNet101, and EfficientNetV2—was compared through training and validation. Among these, the DenseNet201 model demonstrated the highest performance across five performance metrics, achieving an accuracy of 73.0 ± 0.92, precision of 73.2 ± 1.21, recall of 73.0 ± 0.92, and F1-score of 72.7 ± 0.96. The AUROC for DenseNet201 was 94.1 ± 0.78, ranking second only to EfficientNetV2. In terms of qualitative evaluations using Grad-CAM, the DenseNet201 model was also identified as that which could most accurately identify the regions of interest.

Compared to similar existing studies (Table 4), the performance of the K-L grade classification model trained in this study was higher than that of the ResNet50-based classification model developed by Wang et al. in 2020 [10]. This can be attributed to the DenseNet201 model, which demonstrated the best performance in this study, having a deeper and more complex network architecture than the ResNet50 model. This enabled us to extract features in a more refined and diverse manner.

In addition, Cheung et al. (2021) [12] proposed a method that combined a CNN model with XGBoost to predict K-L grades based on joint space width measurements, achieving an AUC of 0.621. In contrast, our best-performing model, DenseNet201, achieved a significantly higher AUC of 0.94, highlighting the advantage of our deep learning-based classification approach trained on balanced clinical data.

Furthermore, as summarized in Table 4, unlike existing studies, in which public datasets such as OAI and MOST have primarily been used, this study utilized clinical data obtained from patients who visited the hospital. This clinical dataset, which reflects the complexity and heterogeneity of real-world medical cases, allows for a more realistic evaluation of model performance. Consequently, the comparative analysis of deep learning architectures on this dataset offers valuable insights into their practical utility for clinical decision-making. Moreover, these data were balanced across each grade, likely contributing to the improved model performance.

All deep learning models showed comparable performance when compared to existing studies, demonstrating that automated evaluation can reduce subjectivity inherent to specialists’ diagnoses. Thus, the use of deep learning models for automated K-L grade classification presents significant advantages over traditional manual K-L grade classification. Furthermore, when compared to the recent study by Lee et al. in 2024, which utilized a plug-in module for automated K-L grade classification of osteoarthritis, both studies faced challenges in accurately classifying K-L Grade 1. However, the model presented in our study demonstrated higher performance in classifying K-L Grade 1. While this difference may be attributed to variations in the datasets used, it suggests that our model holds greater potential for clinical advancement [18].

In addition to performance metrics, we also compared the computational complexity of the three models. DenseNet201 comprised approximately 20.02 million trainable parameters and required 8.62 GFLOPs, making it the most computationally efficient architecture among the three. In contrast, EfficientNetV2 had the largest model size with 118.52 million parameters and 25.90 GFLOPs, whereas ResNet101 had 44.55 million parameters and 16.65 GFLOPs, placing it between the two. These results indicate that DenseNet201 achieved a favorable balance between accuracy and computational efficiency, suggesting that it could be more advantageous for real-world clinical deployment in terms of both performance and resource usage.

In this study, the automated classification of K-L grade was attempted using deep learning models based on regions of interest (ROIs) designated by consensus among two or more specialists. While the models generally showed good performance, there were some noteworthy limitations. As shown in Figure 6, upon visualization for qualitative performance assessment, one noticeable limitation was the fact that the deep learning models did not extract all features from the manually designated ROIs. Additionally, compared to other grades, the models showed lower AUC values for K-L Grade 1, which has the least symptoms of the disease, indicating room for improvement.

Furthermore, the dataset used in this study was collected from a single institution, and the dataset size was relatively small to demonstrate clinical applicability. Therefore, future studies should consider expanding the dataset size and incorporating validation from multiple institutions to enhance the model’s generalizability and robustness. Thus, future studies should focus more on feature extraction during training or employ data augmentation techniques to diversify model training, which is expected to result in more generalized and robust performance.

Additionally, future studies should explore an ensemble model that integrates feature maps extracted from multiple deep learning architectures prior to the classification stage, in order to further enhance feature diversity and model performance. Furthermore, future research will also focus on designing custom deep learning architectures tailored to clinical requirements and exploring advanced ensemble strategies that go beyond feature-level fusion, aiming to improve both interpretability and diagnostic reliability in real-world settings. This approach could potentially address the limitations in feature extraction identified in the current models.

At present, K-L grade classification is typically performed manually based on the subjective judgment of specialists. However, this can lead to significant variability in diagnostic accuracy and reliability depending on the specialist’s level of expertise. In the future, AI-powered technologies that automatically interpret the radiographic results of patients with KOA are expected to reduce the subjectivity associated with the manual approach, thereby enhancing reliability, as well as shortening the time for diagnosis. Taken together, these improvements could greatly benefit patient treatment and management. Particularly, using deep learning models for automatic K-L grade classification could enable radiologists to make more accurate and consistent diagnoses. Moreover, if AI is developed to analyze and quantify the inflamed areas of the knee joint in detail, this AI-assisted diagnostic approach could greatly influence the decision-making process in patient treatment planning.

## Figures and Tables

**Figure 1 diagnostics-15-01220-f001:**
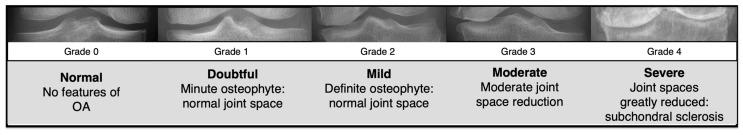
Knee images showing the differences of JSN according to different K-L grade severity.

**Figure 2 diagnostics-15-01220-f002:**
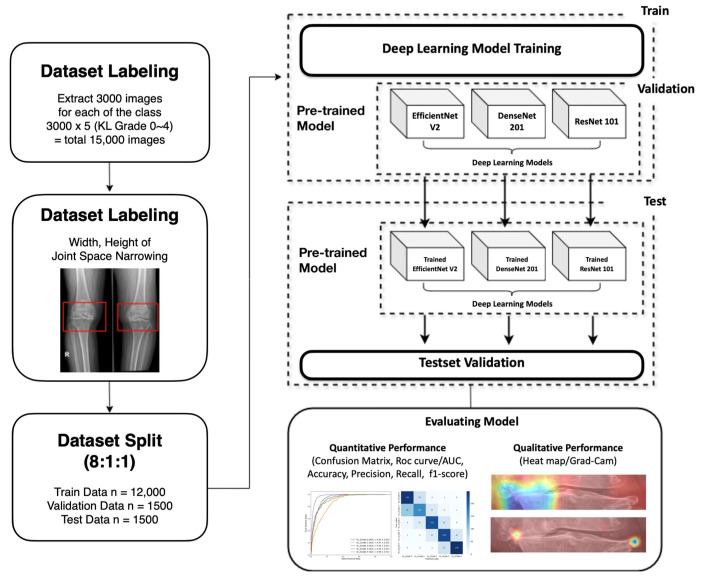
The overall pipeline of the research experiment. A total of 12,000 patient data images were used to train a three-layer CNN and three deep learning models (EfficientNetV2, DenseNet201, and ResNet101), followed by validation and testing. Based on both quantitative and qualitative performance, the model demonstrating the highest performance in K-L grade classification was selected.

**Figure 3 diagnostics-15-01220-f003:**
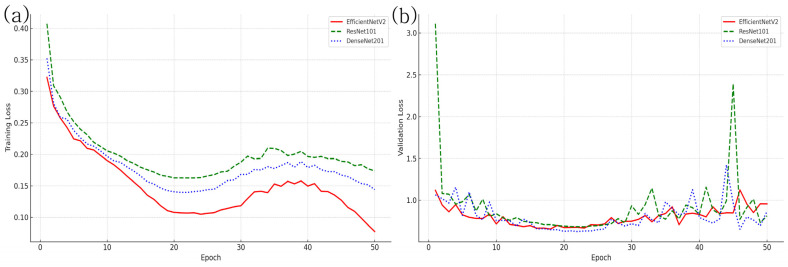
Training and validation loss curves for DenseNet201, ResNet101, and EfficientNetV2 models. (**a**) shows the training loss curves, and (**b**) shows the validation loss curves over 50 epochs. Although a slight overfitting trend was observed in the later epochs for some models, early stopping and checkpoint strategies based on validation loss were used to select the models with optimal generalization performance.

**Figure 4 diagnostics-15-01220-f004:**
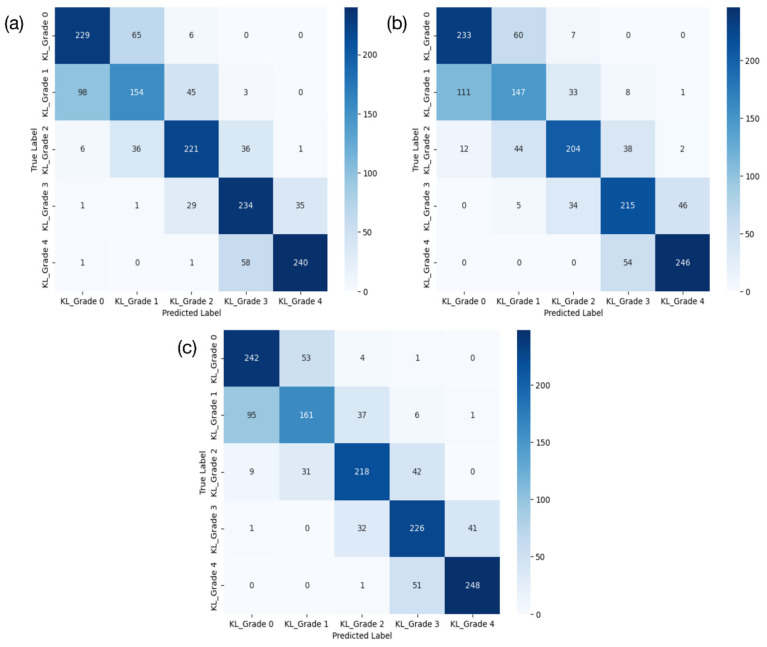
Confusion matrix of classification results. Confusion matrix on EfficientNetV2, ResNet101, and DenseNet201 from left to right (**a**–**c**).

**Figure 5 diagnostics-15-01220-f005:**
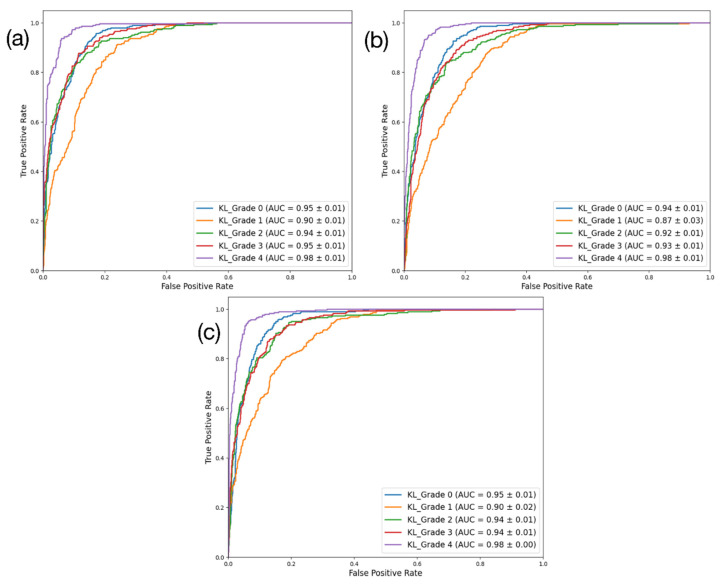
ROC curve and AUC on classification results. EfficientNetV2, ResNet101, and DenseNet201 test results from (**a**–**c**).

**Figure 6 diagnostics-15-01220-f006:**
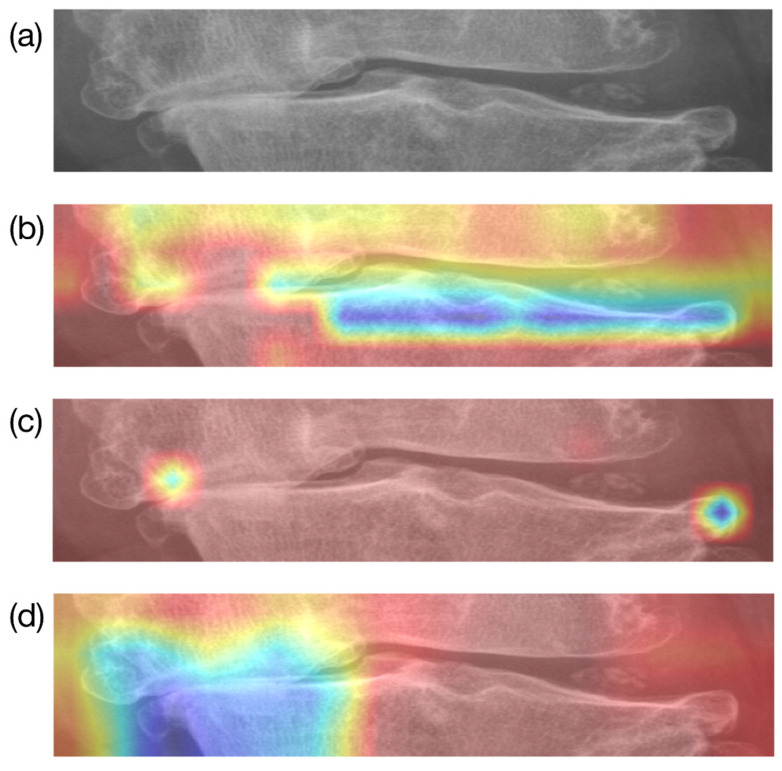
Comparison of attention maps derived from the three deep learning models: (**a**) original image and (**b**–**d**) attention maps derived from EfficientNetV2, ResNet101, and DenseNet201, respectively.

**Table 1 diagnostics-15-01220-t001:** Summary of the performance metrics used for model evaluation.

Metric	Fomula
Accuracy	$\frac{\left(TP\;+\;TN\right)}{\left(TP\;+\;TN\;+\;FP\;+\;FN\right)}\;\;\;$
Precision	$\frac{TP}{\left(TP+FP\right)}\;$
Recall	$\frac{TP}{\left(TP+FN\right)}\;$
F1-score	$2\;\times \;\frac{\left(\mathrm{Pr}ecision\;\times \;Recall\right)}{\left(\mathrm{Pr}ecision\;+\;Recall\right)}\;$
True Positive Rate (TPR)	$\frac{TP}{\left(TP\;+\;FN\right)}\;$
False Positive Rate (FPR)	$\frac{FP}{\left(FP+TN\right)}\;$
AUC (Area Under Curve)	$\int_{0}^{1}TPR\left(FPR\right)d\left(FPR\right)$

The formulas for accuracy, precision, recall (sensitivity), F1-score, True Positive Rate (TPR), False Positive Rate (FPR), and Area Under the Curve (AUC) are provided.

**Table 2 diagnostics-15-01220-t002:** Performance for K-L grade prediction. The reported performances were averaged over three runs. The upward arrow (↑) indicates that a higher value reflects better performance for each metric.

Baseline	Accuracy (%) ↑	Precision (%) ↑	Recall (%) ↑	F1-Score (%) ↑	AUC (%) ↑
EfficientNetV2	71.8 ± 2.18	72.0 ± 2.10	71.8 ± 2.18	71.6 ± 2.14	94.3 ± 0.34
ResNet101	69.6 ± 2.39	69.7 ± 2.51	69.6 ± 2.39	69.3 ± 2.34	92.8 ± 0.83
DenseNet201	73.0 ± 0.92	73.2 ± 1.21	73.0 ± 0.92	72.7 ± 0.96	94.1 ± 0.78

**Table 3 diagnostics-15-01220-t003:** Classification accuracy (%) by K-L grade for each model. The upward arrow (↑) indicates that a higher value reflects better performance.

	K-L Grade 0 (↑)	K-L Grade 1 (↑)	K-L Grade 2 (↑)	K-L Grade 3 (↑)	K-L Grade 4 (↑)
EfficientNetv2	76.33	51.33	73.67	78.00	80.00
ResNet101	77.67	49.00	68.00	71.67	82.00
DenseNet201	80.67	53.67	72.67	75.33	82.67

**Table 4 diagnostics-15-01220-t004:** Literature summary. The upward arrow (↑) indicates that a higher value reflects better performance for each metric.

Study	Model	Dataset	Metric	Performance (↑)
Tiulpin et al. (2018) [8]	ResNet 34	MOST/OAI	Accuracy	>66%
Antony et al. (2019) [9]	CNN Fine-tuned	MOST/OAI	Accuracy	>57%
Wang et al. (2020) [10]	ResNet50	OAI	Accuracy	≈69.18%
Olsson et al. (2021) [11]	CNN	Clinical Data	Mean AUC	0.92
Cheung et al. (2021) [12]	CNN + XGBoost(JSW)	OAI	AUC	0.621
Chern et al. (2022) [13]	DenseNet201 + SVM	OAI (balanced)	AUC	0.71
Lee et al. (2024) [18]	Pulg-in Module	OAI + MOST	Accuracy	43% (Grade 1)
Our Study	DeseNet201	Clinical Data (balanced)	Accuracy	73%
Our Study	DeseNet201	Clinical Data (balanced)	AUC	0.94
Our Study	DeseNet	Clinical Data (balanced)	Accuracy	53.67% (Grade 1)

## Data Availability

The datasets generated and/or analyzed during the current study are available from the corresponding author on reasonable request. The code used during the current study is available from the corresponding author on reasonable request.

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
