# Peer review of "Artificial Intelligence Model Assists Knee Osteoarthritis Diagnosis via Determination of K-L Grade"

_diagnostics, 2025, doi:10.3390/diagnostics15101220_

Round 1

Reviewer 1 Report

Comments and Suggestions for Authors

The article determines the evaluation of KOA severity according to the Kellgren-Lawrence rating system using deep learning models. In the study where a total of 15000 images were used, DenseNet201, ResNet101, and EfficientNetV2 architectures were used. The overall accuracy was 73%. The article is generally well prepared. The use of attention maps and medical explanations especially adds a separate value to the article. However, it would be appropriate to add a few of my suggestions to the article.

1. Considering the complex structures of medical images, using architectures such as Densenet and Resnet are suitable approaches. The authors made the right choice. The reason for this choice should be emphasized.

2. Obtaining the confusion matrices is not enough to show the effectiveness of these architectures on the dataset. Add the loss function change for the Densenet architecture, where the highest accuracy value is obtained. This change indicates whether the training process is complete or not.

3. Add a table showing the literature summary to the discussion section. Relate the discussion you made on page 9 to the table.

4. Justify why you did not do cross-validation. It should have been done to eliminate or show that there was no overfitting problem.

5. In the limitation paragraph, it is stated that not all features can be obtained. A solution for this is to combine the feature maps obtained from all architectures and use this feature map in the classification process. This solution may eliminate the described limitation.

6. Section 2 should have a methods section. This section should have the following subheadings.

Experimental environment
K-L grade classification models
Models evaluation
7. In the Models evaluation section, add the equations of the performance parameters in a table.

Reviewer 2 Report

Comments and Suggestions for Authors

This manuscript utilized DenseNet201, ResNet101, and EfficientNetV2 deep learning models  to automate the assessment of KOA severity using the Kellgren-Lawrence grading system (class 0~4). The performance of these three models in lesion classification was evaluated and compared, using the evaluation metrics of accuracy, precision, recall, and F1-Score. A total of 15,000 images were used, with 3,000 images collected for each grade.

My concerns are as follows:

  1. The contributions of these manuscript should be listed in the Introduction section.
  2. The novelty of this manuscript is limited. In terms of the deep learning models, this manuscript only used the existing models, but lacks improvements and innovative approaches to the models. In terms of the analysis of the medical experimental results, there is no integration of clinical perspectives to interpret the experimental findings.
  3. Authors should give the formulas of the evaluation metrics Accuracy, Precision, Recall, F1-Score, Receiver Operating Characteristic (ROC) Curve, and Area Under the Curve (AUC).
  4. Authors need to improve the Results sectionto conduct comparative studies against best performing methods published by other researchers in reputable sources (i.e. high impact factor journals and/or high tier conferences) for solving the same problem (if non-existing, most similar).
  5. In the tables ofresults, authors need to include up-arrow or down-arrow next to each evaluation metrics, to show whether higher or lower value is better.
  6. Authors cancompare/comment on the computational complexity of the compared methods.
  7. As described in the last paragraph of the Introduction section, the dataset with balanced data across K-L grades is a contribution of this manuscript, so the dataset should be made publiclyavailab

Round 2

Reviewer 2 Report

Comments and Suggestions for Authors

Thank you for the revised version.

The authors have improved the manuscript based on all the suggestions, such as adding the contribution of the manuscript, the discussion of clinical insight, the formulas of evaluation criteria, the experimental comparison with other methods, and etc.